# Exploring the Association of the COVID-19 Pandemic on Employee Sleep Quality at a Healthcare Technology and Services Organization

Daniel Arku [1], Jennifer M. Bingham [1,2], Jacques Turgeon [3,4], Veronique Michaud [3,4], Terri Warholak [1] and David R. Axon [1,*]

1 College of Pharmacy, University of Arizona, Tucson, AZ 85271, USA; arku@pharmacy.arizona.edu (D.A.); jbingham@trhc.com (J.M.B.); warholak@pharmacy.arizona.edu (T.W.)
2 Tabula Rasa HealthCare Group, Office of Translational Research and Residency Programs, Moorestown, NJ 08057, USA
3 Tabula Rasa HealthCare Group, Precision Pharmacotherapy Research & Development Institute, Orlando, FL 32827, USA; jturgeon@trhc.com (J.T.); vmichaud@trhc.com (V.M.)
4 Faculty of Pharmacy, Université de Montréal, Montréal, QC H3T 1J4, Canada
* Correspondence: axon@pharmacy.arizona.edu; Tel.: +1-520-621-5961

**Abstract:** The COVID-19 pandemic led to global healthcare consequences including insomnia. This survey used the Pittsburgh Sleep Quality Index (PSQI) to assess sleep quality at two time points (July 2020 and November 2020) among employees at a healthcare technology and services organization during the COVID-19 pandemic. Of the 1280 eligible employees, 251 complete responses (response rate, RR = 19.6%) in July and 108 (RR = 8.4%) in November were received and analyzed. The overall mean global PSQI scores were 7.3 ± 3.6 in July and 7.7 ± 3.6 in November 2020 ($p > 0.05$). There was no significant difference in any of the PSQI components or global scores between periods. Our findings indicate poor reported sleep quality among our study participants during the COVID-19 pandemic. Additional studies are needed to assess the longitudinal impact on sleep quality post-COVID-19 pandemic.

**Keywords:** COVID-19; sleep quality; quarantine; PSQI



## 1. Introduction

In late December 2019, initial cases of an unfamiliar cause of pneumonia were reported in Wuhan, Hubei Province, China [1]. These cases were subsequently defined as coronavirus disease-2019 (COVID-19) [2–4]. COVID-19 was declared a pandemic on 11 March 2020, following a rapid, exponential increase in global cases [5]. As of 29 December 2021, over 280 million global confirmed cases and over 5 million confirmed COVID-19 deaths have been reported, with over 8 billion administered COVID-19 vaccinations [6].

The various quarantine, lock-down, and stay-at-home requirements enacted by international governments to control the spread of COVID-19 resulted in unpleasant psychological experiences [7]. These experiences were postulated to be caused by a feeling of constrained freedom and the absence of effective prophylactic regimens or treatments [7]. Specifically, the COVID-19 pandemic was associated with depression, anxiety, sleep disturbances, distress, and suicidal ideations [8–11]. Similar experiences were observed in populations afflicted by other epidemics, due in part to feelings of fear and helplessness [7,12–16].

Huang et al. [17] observed the mental health burden on healthcare workers in China during the early stages of the COVID-19 pandemic. Healthcare workers with increased work-related stress were at risk for sleep disorders and reduced sleep quality [17,18]. Qiu et al. [19] and Qi et al. [20] indicated a higher prevalence of sleep disturbances in Chinese medical workers compared to the general population. High-quality sleep is

essential to good physical health, as it promotes an optimal immune system and reduces susceptibility to infection [21].

However, little is known about the association between the COVID-19 pandemic and sleep quality among United States (US) employees at healthcare technology and services organizations. The aim of this pilot study was to assess sleep quality among employees at a healthcare technology and services organization in the US throughout the COVID-19 pandemic.

## 2. Materials and Methods

### 2.1. Study Design and Instruments

This project employed the Pittsburgh Sleep Quality Index (PSQI) [22] to capture participants' sleep quality at two time points during the COVID-19 pandemic: (1) July 2020; and (2) November 2020. Both periods were chosen to closely reflect two different seasons and changes of events during the pandemic; during the summer of 2020 when people were mostly active with cases increasing [6] and during the fall of 2020 when stay-at-home orders were enacted before the holidays in the US. The PSQI included an 18-item scale and assessed seven specific components, including: (1) sleep quality, (2) sleep duration, (3) sleep latency, (4) habitual sleep efficiency, (5) sleep disturbance, (6) use of sleeping medications, and (7) daytime dysfunction [22,23]. In accordance with the PSQI scoring instructions, each item was scored between 0 and 3 [22]. The sum of these scores ranged from 0–21 and represented a global score for each participant, where a higher score indicated poorer sleep quality [22]. Participants were also asked to report their age, gender, professional credentials, current area of pharmacy practice, whether they had been exposed or knew someone who had been exposed to COVID-19, whether their professional role involved ordering or administering COVID-19 tests, and whether they interacted with patients face-to-face, via telehealth, and/or via telephone in their current position.

The questionnaire was administered through a Research Electronic Data Capture (REDCap) secure web-based survey (Vanderbilt University Medical Center, Version 8.10.0, Nashville, TN, USA) [REDCap]. The University of Arizona Institutional Review Board approved this project (protocol number 2006748396, approved 17 July 2020).

### 2.2. Project Participants

All employees aged 18 years or older were invited to participate in the surveys. There were no exclusion criteria specific to occupation; however, incomplete questionnaire responses were excluded from analyses. Employees encompassed a wide variety of professions, including pharmacists, student pharmacists, pharmacy technicians, information technologists, nurses, student nurses, and lawyers.

### 2.3. Data Collection

Data were collected anonymously over 2-week periods in July and November 2020. All eligible employees ($n = 1280$) received an initial recruitment email inviting them to participate in this project. The email contained information about the project and a REDCap survey link. A reminder email was distributed to all employees one week after the initial email, and the survey closed after one additional week.

### 2.4. Statistical Analysis

In the analysis, the mean component and mean global sleep scores were compared using a one sample t-test. An alpha level of 0.05 was set a priori, and a Bonferroni correction applied for multiple comparisons. Hence, the alpha level for the component and global scores was 0.006. Analyses were conducted using the SAS Studio University Edition (SAS institute Inc., Cary, NC, USA).

### 3. Results

The survey was distributed to 1280 participants employed by the nationwide health-care organization in July and November of 2020. In July 2020, a total of 266 (21.0% response rate (RR)) responses were recorded. However, 15 responses were excluded due to incomplete data that prevented calculation of the PSQI score. Thus, 251 (19.6% RR) responses were included in the analysis. Three months later, in November 2020, a total of 113 (8.8% RR) responses were recorded. However, five responses were excluded due to incomplete data that prevented calculation of the PSQI score; thus, 108 (8.4% RR) responses were included in the analysis.

Participants were of similar ages in both the July (41.8 ± 12.5 years) and November 2020 (41.7 ± 12.8 years) groups ($p > 0.05$). Most participants were female (July = 68.3%, November = 69.8%) and reported not knowingly being exposed to or having known someone with COVID-19 (July = 59.0%, November = 50.9%). Very few participants (~1.0%) reported ordering or administering COVID-19 laboratory diagnostic tests. See Table 1.

**Table 1.** Demographic characteristics of project participants by data collection date.

| Variable | July 2020 $n = 251$ | November 2020 $n = 108$ |
|---|---|---|
| Age, mean ± SD years | 41.8 ± 12.5 | 41.7 ± 12.8 |
| Gender, N (%) | | |
| Male | 78 (31.7) | 32 (30.2) |
| Female | 168 (68.3) | 74 (69.8) |
| Exposed or know someone exposed to COVID-19, N (%) | | |
| Yes | 103 (41.0) | 53 (49.1) |
| No | 148 (59.0) | 55 (50.9) |
| Professional role involves ordering or administering COVID-19 lab tests, N (%) | | |
| Yes | 3 (1.2) | 1 (0.9) |
| No | 248 (98.8) | 106 (99.1) |

NA = data were not collected. SD = standard deviation. PSQI = Pittsburgh Sleep Quality Index.

There was no statistically significant difference in mean global PSQI scores and component scores between data collected in the July and November surveys (global scores: 7.3 ± 3.6 versus 7.7 ± 3.6, $p = 0.4355$, Bonferroni adjusted $p > 0.006$). See Table 2.

**Table 2.** Component and global scores on the Pittsburgh Sleep Quality Index (PSQI) for project participants by data collection date.

| PSQI Component | July 2020 ($n = 251$) Mean ± SD | November 2020 ($n = 108$) Mean ± SD | $p$ |
|---|---|---|---|
| Subjective sleep quality | 1.2 ± 0.7 | 1.2 ± 0.7 | 0.9814 |
| Sleep latency | 1.4 ± 0.8 | 1.5 ± 0.9 | 0.2888 |
| Sleep duration | 1.0 ± 0.9 | 1.0 ± 0.8 | 0.7707 |
| Habitual sleep efficiency | 0.8 ± 1.1 | 0.6 ± 0.9 | 0.0827 |
| Sleep disturbances | 1.4 ± 0.6 | 1.5 ± 0.6 | 0.2128 |
| Use of sleeping medication | 0.7 ± 1.1 | 0.9 ± 1.3 | 0.1126 |
| Daytime dysfunction | 0.9 ± 0.7 | 1.1 ± 0.8 | 0.0752 |
| Global | 7.3 ± 3.6 | 7.7 ± 3.6 | 0.4355 |

SD = standard deviation. Alpha level after Bonferroni correction = 0.006. In accordance with the PSQI scoring instructions, each item was scored between 0 and 3 and summed to obtain a global PSQI global score with a possible range of 0–21, with higher scores indicating poor sleep quality. A score of >5 indicated poor sleep quality.

## 4. Discussion

Findings from this pilot study indicate poor reported sleep quality among employees at one national healthcare technology and services organization in the US at two time points during the COVID-19 pandemic in 2020. These findings align with other international studies that associate the COVID-19 pandemic with worsened sleep quality [24,25]. Further, the data collected, using the PSQI, showed no statistically significant difference in sleep quality among employees at the same organization between July and November 2020.

The mean global PSQI score of 8.61 also indicated poor sleep quality [22] and was comparatively greater than PSQI scores reported elsewhere. For example, one study in China reported a mean PSQI score of 4.88 ± 2.96 among the non-diseased public [24]. In another example, the mean PSQI score among Chinese residents was 4.85 ± 3.11 [25]. However, differences between these studies may be explained by different participant populations and differences between the US and Chinese healthcare systems.

This finding suggests poor sleep quality due to COVID-19 may not be limited to those with direct patient contact in healthcare settings, but can affect all employees. Some of these employees might be involved in taskforce committees that are responsible for planning, implementing, and enforcing workplace policies on COVID-19. Hence, this may be associated with their sleep quality. However, other studies observed that the influence of stress on sleep quality, especially among front line employees or employees in patient-facing roles, was higher than in the non-disease public, due to the perceived increased risk of infection among healthcare employees [19,20].

Data from the PSQI found no statistically significant difference in sleep quality between July and November of 2020, which indicates that poor reported sleep quality persisted during the peak of the pandemic in 2020. There may be several explanations for the poor sleep quality observed during the peak of the pandemic in the study. In particular, the US and other countries enacted restrictions to help reduce the spread of this virus [26,27]. Stay-at-home and quarantine orders likely necessitated changes in people's routines, including the need to: create a makeshift home office environment, deal with work disruptions and distractions, manage childcare and other caring responsibilities, and manage stress from uncertainty about life and insecurity about their health [26–28]. The COVID-19 pandemic may have impeded normal sleep patterns due to its impact on anxiety and depression [24,29]. With the high prevalence of insomnia symptoms associated with frequent wakefulness and early awakening during sleep [30], such findings support the notion that the outbreak may have contributed to increasing cases of low sleep quality among the public.

In this project, sleep latency, which measures the time to fall asleep and difficulty to get to sleep within 30 min [22], was among the highest PSQI component scores in both periods. Likewise, sleep disturbances (e.g., waking up in the middle of the night or early morning, or feeling too cold or too hot) [22] was also among the highest PSQI component scores observed in both periods. In our project, most participants reported waking up three or more times per week in the middle of the night or early morning and had difficulty sleeping due to feeling too hot at night. During high temperatures, the body temperature decreases, leading to an increase of the average heart rate and respiratory rate, thus increasing wakefulness [31,32].

To the best of our knowledge, this is the first pilot study of its kind to assess sleep quality in response to the COVID-19 pandemic in the US and adds to the growing interest in the health consequences of COVID-19. To date, efforts to eradicate the virus are projected to continue, given public health initiatives. Further research is needed to assess sleep quality beyond the pandemic recovery phase.

This project had some limitations. First, the cross-sectional measures used in the PSQI only account for responses at one point in time and cannot infer causality. Thus, it may be difficult to identify issues that change frequently, given the unprecedented and unpredictable nature of the COVID-19 pandemic. Sleep quality was only captured among employees at one organization and had a low response rate, even though all

eligible participants were invited to participate; thus, the findings may not be generalizable beyond this population. This project did not capture potential confounding risk factors for sleep quality, such as stress, underlying health conditions, anxiety, and depression [11,26]. Data were self-reported rather than obtained from objective assessments (e.g., through polysomnography or actigraphy), and thus sleep quality may have been overestimated or underestimated [33,34]. Additionally, some participants may have a history of frequent poor sleep quality and others may have misclassified their sleep quality due to difficulty in recalling, which could underestimate or overestimate the association. However, if the recall bias is assumed to affect both groups equally, then the effect should be muted.

Future research should adopt a longitudinal study design, recruit a larger and more representative sample, and adjust for potential confounding variables to better assess the impact of COVID-19 on sleep quality. Future research could also evaluate sleep quality in an additional group of healthcare workers who experienced COVID-19 to see how the results compare to the findings of the current study.

### 5. Conclusions

Employees of one healthcare technology and services organization in the US reported poor sleep quality at two time points during the COVID-19 pandemic. There was no statistically significant difference in perceived sleep quality among participants in the study between the two time periods assessed. Subsequent studies should measure anxiety and other stress indicators using experimental or longitudinal designs to better assess the impact of COVID-19 on perceived sleep quality post-pandemic.

**Author Contributions:** Conceptualization, J.M.B., D.R.A. and D.A.; methodology, T.W., D.R.A. and D.A.; software, D.A.; validation, D.R.A., J.M.B., T.W. and J.T.; formal analysis, D.A. and D.R.A.; data curation, D.A. and D.R.A.; writing—original draft preparation, D.A.; writing—review and editing, D.A., D.R.A., J.M.B., T.W. and J.T.; visualization, D.A.; supervision, D.R.A., T.W., J.T. and V.M.; project administration, J.M.B.; funding acquisition, T.W. All authors have read and agreed to the published version of the manuscript.

**Funding:** This research received an unrestricted grant from Tabula Rasa HealthCare Group.

**Institutional Review Board Statement:** The study was conducted according to the guidelines of the Declaration of Helsinki and approved by the Institutional Review Board (or Ethics Committee) of the University of Arizona (protocol code 2006748396, approved 17 July 2020).

**Informed Consent Statement:** Informed consent was obtained from all subjects involved in the study.

**Data Availability Statement:** The data presented in this study are available on request from the corresponding author. The data are not publicly available, to protect participant confidentiality.

**Acknowledgments:** We would like to thank Niloufar Emami, for her assistance with the project.

**Conflicts of Interest:** The authors declare no conflict of interest.

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
