# Peer review of "Exploring the Association of the COVID-19 Pandemic on Employee Sleep Quality at a Healthcare Technology and Services Organization"

_covid, doi:10.3390/covid2020012_

Round 1
Reviewer 1 Report
Accept
Reviewer 2 Report
The revised version of the manuscript is now more detailed and complete. The authors well-addressed all of my previous comments. I have no additional comments on this paper. It can be accepted for publication after the editorial check.
This manuscript is a resubmission of an earlier submission. The following is a list of the peer review reports and author responses from that submission.
Round 1
Reviewer 1 Report
Arku and colleagues presented a study aimed at evaluating the quality of sleep before and during pandemic as well as during the vaccination in the US. The manuscript is very interesting and gives novel information about the psychological impact of the COVID-19 pandemic in healthcare workers. Overall, the manuscript is well written, below are reported some minor comments that will improve the quality of the manuscript:
1) The epidemiological data of the following sentence should be updated: “As of July 30, 2021, over 196 million global confirmed cases and over four million confirmed COVID-19 deaths have been reported, with over three billion administered COVID-19 vaccinations [6].”;
2) It would have been interesting to evaluate the sleep quality in an additional group of healthcare workers who experienced COVID-19 infection. I know that this aspect is out of the scope of the study, but the inclusion of this further group would be interesting;
3) It is not clear why the authors chose July 2020 and November 2020 as the study period of the second part of the analysis;
4) Please consider to shorten the Discussion section.
Reviewer 2 Report
I have just received the ms entitled “Exploring association of the COVID-19 Pandemic and Sleep Quality of Employees at a Healthcare Technology and Services Organization”.
In my opinion, the paper presents several flaws that do not make suitable for publication.
The first important issue regards the huge confusion between SARS-CoV-2 and COVID-19: the latter is the disease associated with the first, which is the viral agent. The difference is that the first can by asymptomatic; where symptoms appear it becomes COVID-19.
Authors erroneously wrote:
- l. 15: The SARS-CoV-2 (COVID-19): SARS-CoV-2 (COVID-19)
- l. 37: novel severe acute respiratory syndrome coronavirus 2 (SARS-CoV-2) [2-3], later referred to by the World Health Organization (WHO) as coronavirus disease-2019 (COVID-19): SARS-CoV-2 was not later referred as coronavirus disease 2019; once again, the COVID-19 is the disease due to infection with SARS-CoV-2
I have doubts on the nature of the study: authors considered it as a “quality improvement project”; in my opinion, this is a survey-based cross-sectional study.
- 57. Consequently, healthcare workers who are exposed to environmental stress are at increased risk of infection. I’d like to see clarifications on this point. Few evidence has suggested the possible that stress-related condition may increase the risk of acquiring certain infections, but this sentence is confusing. Do authors refer to the risk of being infected with SARS-CoV-2? are there any evidence?
The study relies on a 3-item questionnaire: I have doubt that this is enough to reach firm conclusions on the definition of sleep disorders and COVID-19. As known, mental disorders and symptoms need robust instruments for reach a strong and reliable diagnosis. In particular, due to the scarcity of items, respondents could be influenced by thinking at researchers’ expectations. Again, the dependent and independent variables cannot be verified.
From a methodological standpoint, there is no sample size calculation and the response rate is very low, limiting the real-world potential of this investigation. In this regard, once amended all the other issue, the paper could be presented as a pilot study.
Among limitations, authors recognized possible bias due to self-reporting. But “thus sleep quality may have been overestimated or underestimated” is not enough in this regard. Authors might want to add possible bias that have may lead to those, also considering the general COVID-19 environment that may have led to choose a specific answer, affecting the ability to
establish the directionality in the investigated relationship
Round 2
Reviewer 2 Report
Dear Authors,
I understand the efforts made to improve your paper, but I think that it deserves more deep investigation before considering it for publication. However, the responses cannot be considered satisfactory.
For instance,
In particular, a sample size should be not considered "reasonable", but representative of the target population.
"Thank you for the clarification, we have revised the text and used the term COVID19 throughout for simplicity as that is the term most people are familiar with." scientific paper does not rely on familiarity with terms. The use of terms should be appropriate to the scientific purposes of the paper.